# Relationship between Excreted Uremic Toxins and Degree of Disorder of Children with ASD

**DOI:** 10.3390/ijms24087078

**Published:** 2023-04-11

**Authors:** Joško Osredkar, Barbara Žvar Baškovič, Petra Finderle, Barbara Bobrowska-Korczak, Paulina Gątarek, Angelina Rosiak, Joanna Giebułtowicz, Maja Jekovec Vrhovšek, Joanna Kałużna-Czaplińska

**Affiliations:** 1Institute of Clinical Chemistry and Biochemistry, University Medical Center Ljubljana, Njegoseva 4, 1000 Ljubljana, Slovenia; barbara.zvar-baskovic@nib.si (B.Ž.B.); petra.finderle@kclj.si (P.F.); 2Faculty of Pharmacy, University of Ljubljana, Aškerčeva Cesta 7, 1000 Ljubljana, Slovenia; 3Department of Toxicology and Food Science, Faculty of Pharmacy with the Laboratory Medicine Division, Medical University of Warsaw, Banacha 1, 02-097 Warsaw, Poland; barbara.bobrowska@wum.edu.pl; 4Institute of General and Ecological Chemistry, Faculty of Chemistry, Lodz University of Technology, Zeromskiego 116, 90-924 Lodz, Poland; paulina.gatarek@p.lodz.pl (P.G.); angelina.rosiak@p.lodz.pl (A.R.); 5CONEM Poland Chemistry and Nutrition Research Group, Lodz University of Technology, Zeromskiego 116, 90-924 Lodz, Poland; 6Department of Bioanalysis and Drug Analysis, Faculty of Pharmacy with the Laboratory Medicine Division, Medical University of Warsaw, Banacha 1, 02-097 Warsaw, Poland; joanna.giebultowicz@wum.edu.pl; 7Center for Autism, Unit of Child Psychiatry, University Children’s Hospital, University Medical Centre Ljubljana, Zaloška c.002, 1000 Ljubljana, Slovenia; maja.jekovec@guest.arnes.si

**Keywords:** uremic toxins, ASD, degree of deficit

## Abstract

Autism spectrum disorder (ASD) is a complex developmental disorder in which communication and behavior are affected. A number of studies have investigated potential biomarkers, including uremic toxins. The aim of our study was to determine uremic toxins in the urine of children with ASD (143) and compare the results with healthy children (48). Uremic toxins were determined with a validated high-performance liquid chromatography coupled to mass spectrometry (LC-MS/MS) method. We observed higher levels of p-cresyl sulphate (pCS) and indoxyl sulphate (IS) in the ASD group compared to the controls. Moreover, the toxin levels of trimethylamine N-oxide (TMAO), symmetric dimethylarginine (SDMA), and asymmetric dimethylarginine (ADMA) were lower in ASD patients. Similarly, for pCS and IS in children classified, according to the intensity of their symptoms, into mild, moderate, and severe, elevated levels of these compounds were observed. For mild severity of the disorder, elevated levels of TMAO and comparable levels of SDMA and ADMA for ASD children as compared to the controls were observed in the urine. For moderate severity of ASD, significantly elevated levels of TMAO but reduced levels of SDMA and ADMA were observed in the urine of ASD children as compared to the controls. When the results obtained for severe ASD severity were considered, reduced levels of TMAO and comparable levels of SDMA and ADMA were observed in ASD children.

## 1. Introduction

Autism spectrum disorder (ASD) includes complex developmental disorders that affect communication and behavior and symptoms that include persistent deficits in social communication, social interaction, and repetitive, restricted patterns of behavior, interests, and activities [1]. According to Centers for Disease Control and Prevention estimates, ASD occurs in one in fifty-four children by the age of eight and is four times more common among boys than among girls; moreover, there is no evidence of differences in prevalence between races [2]. Even now, the exact cause of ASD is not known. Many researchers believe that the symptoms on which ASD is diagnosed are the result of a number of factors, both environmental and genetic [3]. About 10–25% of ASD cases are explained by mutations at specific genetic loci. However, studies of twins have suggested that genetic and environmental factors equally affect the risk of ASD [4]. Identification of environmental factors involved in development of ASD is therefore crucial for a better understanding of its etiology. There is increasing evidence of gut microbiota–brain axis involvement in ASD, related to gastrointestinal dysfunction manifested by increased intestinal permeability, as well as dysbiosis of gut microbiota [5,6,7]. Studies showed that patients with ASD are 4.4 times more likely to experience gastrointestinal symptoms than are neurotypicals [8]. The most common symptoms are abdominal pain, diarrhea, constipation, and abdominal bloating [9]. ASD may also be associated with intestinal inflammation, which may eventually become systemic [10]. Both gastrointestinal problems and immune dysfunction in patients with ASD may be closely linked to microbiota composition abnormalities [10].

The above factors are all potential causes of elevated blood uremic toxin (UT) concentrations and consequent increased urinary excretion in children with ASD. For example, chronic constipation is a commonly described problem in children with ASD, leading to increased intestinal permeability, which would result in elevated UT concentrations in the blood of these children. However, as UTs are excreted by the kidneys, these elevated concentrations would also be observed in the urine [11].

Many compounds that belong to uremic toxins are directly related to gut dysbiosis. The bacterial origins of compounds such as p-cresyl sulphate (pCS), indoxyl sulphate (IS), indole-3-acetic acid, trimethylamine (TMA), trimethylamine N-oxide (TMAO), hippuric acid, phenol, and phenylacetic acid were proven in several independent experiments [12,13,14,15].

The source of circulating TMAO is the precursor TMA, produced by the gut microbiome from metabolism of dietary choline. Both TMAO and TMA are well-known water-soluble uremic toxins and are considered to be related to the microbiome [16]. TMAO crosses the blood–brain barrier, which makes it a candidate for influencing cognitive function and neurological disease [17,18,19,20,21].

Urinary asymmetric dimethylarginine (ADMA) and symmetric dimethylarginine (SDMA) belong to the guanidine group. They are small, water-soluble molecules that are L-arginine analogues [22]. Recent studies have suggested that guanidines are also involved in uremic complications such as inflammation, inhibition of endothelial cell proliferation, and reduction in osteoblastic calcification [23,24]. SDMA is an isomer of ADMA and for a long time was considered biologically inactive. However, studies have suggested that SDMA decreases nitric oxide (NO) synthesis in endothelial cells while increasing reactive oxygen species (ROS) synthesis. SDMA also indirectly promotes inflammation as well as endothelial damage. In contrast to ADMA, it is not reduced in enzymatic concentration, but concentration of SDMA depends mainly on renal excretion from the body [23,25].

The aryl sulphates pCS and IS are protein-bound UTs. Early studies suggested that IS is a metabolite of tryptophan (Trp). Trp, consumed with food, is converted to indole in the intestine by local bacteria and then absorbed into the systemic circulation. Indole is sequentially metabolized in the liver into indoxyl and IS, the latter of which is then excreted by the kidneys [26]. IS is also related to cardiovascular disease in patients with chronic kidney disease through induction of oxidative stress in endothelial cells [27], and peripheral IS levels are related to reduced cognitive function in renal dialysis patients [28]. IS can cross the blood–brain barrier [29]. Elevated IS levels have been observed in patients with ASD [30] and Parkinson’s disease [31], suggesting that it is also associated with neurodevelopmental and neurodegenerative diseases. IS can be considered a neurotoxin for its ability to accumulate in brain tissue, especially as a result of disruption of organic anion transporter 3 (OAT 3) function at the blood–brain barrier [32]. Significantly, IS may contribute to disrupted central-nervous-system homeostasis and neuronal damage directly through increased oxidative stress and inflammation in glial cells [33].

Indoles could also have important immunomodulatory effects and are strong agonists for aryl hydrocarbon receptors (AHRs) [34], which regulate host immunity and barrier function at mucosal sites [35].

p-Cresol, a small aromatic metabolite, is one of the microbial metabolites associated with ASD for which results are more consistent in various studies. According to four independent studies, elevated urinary [36,37] and fecal [38,39] p-cresol levels were found in patients with ASD. Urinary p-cresol levels also correlated with severity of behavioral changes in ASD [36].

The aim of this study was to investigate whether elevated levels of uremic toxins (pCS, IS, TMAO, ADMA, SDMA) in children’s urine samples could be a useful tool for ASD diagnosis. Particular attention was also focused on determination of whether concentration of excreted uremic toxins depends on the degree of deficit of children with ASD and the relationship between uremic toxin concentrations and additional diagnoses.

## 2. Results

Average measurements were used to show the concentration of each toxin in a group of all patients (143 vs. 48). The comparison of TMAO, ADMA, SDMA, IS, and pCS urine levels between ASD children and the control group are presented in Table 1 and Figure 1. The results evidence a median value of the urine levels of IS and pCS that was significantly higher in the ASD sample compared to the controls; *p* = 0.0083 and *p* = 0.0182, respectively. Lower median concentrations were observed in the group of ASD children as compared to the controls, but these differences were not statistically significant (TMAO, *p* = 0.2400; SDMA, *p* = 0.1990; and ADMA, *p* = 0.1820).

We calculated the values of the 2.5th and 97.5th percentiles, with 90% confidence intervals for the lower and upper limits, as recommended by the International Federation of Clinical Chemistry and Laboratory Medicine (CLSI C28-A3). The results are presented in Table 2.

There were no statistically significant differences between the study groups (ASD vs. control) based on gender (Appendix A). In a further step, it was checked whether uremic toxin levels were dependent on the ages of the patients included in this study. We divided the group into those younger than 8 years and those older than 8 years. In the group of children under 8 years of age, there were 46 participants, while in the group of children over 8 years, there were 97 subjects. In all cases, there were no statistically significant differences (TMAO, *p* = 0.0829; SDMA, *p* = 0.6410; ADMA, *p* = 0.9320; pCS, *p* = 0.6800; and IS, *p* = 0.3960) (Appendix A). Average measurements were used to show the concentration of each uremic toxin in a group of patients with additional diagnoses (epilepsy and recurrent seizures; other and unspecified hearing loss; convulsions, not elsewhere classified; delayed milestones in childhood; specific developmental disorders; attention-deficit hyperactivity disorder; tic disorder (additional diagnosis group of 86 vs. control group of 48)). Statistically significant differences were observed for pCS (*p* = 0.0072) and IS (*p* = 0.0182), while no statistically significant differences were found for TMAO (*p* = 0.2570), SDMA (*p* = 0.4060), or ADMA (*p* = 0.5580) (Figure 2).

Average measurements were also used to show the concentration of each uremic toxin in a group of ASD-only patients as compared to the controls (105 vs. 48). Statistically significant differences were observed for pCS (*p* = 0.0133) and IS (*p* = 0.0234), while no statistically significant differences were found for TMAO (*p* = 0.1520), SDMA (*p* = 0.9570), or ADMA (*p* = 0.9750) (Appendix A). Additionally, in a group of ASD patients with additional diagnoses as compared to the controls (75 vs. 48), statistically significant differences were observed for pCS (*p* = 0.0110) and IS (*p* = 0.0252), while no such relationship was observed for the other compounds determined in this study (*p* > 0.05) (Appendix A). We also compared the obtained levels of the determined uremic toxins in the group of subjects with HFA diagnoses to the control group (38 vs. 48). Statistically significant differences were observed for TMAO (*p* = 1.19 × 10^−2^), SDMA (*p* = 3.82 × 10^−10^), and ADMA (*p* = 1.69 × 10^−9^), and in HFA patients with additional diagnoses, compared to the controls (11 vs. 48), statistically significant results were found for SDMA (*p* = 7.27 × 10^−5^) and ADMA (*p* = 1.04 × 10^−4^) (Appendix A). Figure 3 presents the interactions between the different classification levels of severity (mild, moderate, and severe) and the urinary metabolite levels determined as compared to the control group.

Statistical significance differences were observed for pCS (*p* = 0.0263) and IS (*p* = 0.0264) in the mild-classified group and for SDMA (*p* = 0.0037) and ADMA (*p* = 0.0104) in the moderate-classified group, while no such relationship for any of the determined compounds was observed in the severe-classified group (*p* > 0.05). A question arises as to whether quantification of these compounds provides identical diagnostic information or whether it is a complementary source of information about pathological effects produced via different biological pathways. Until now, mainly, toxic action of single solutes has been emphasized, without consideration of interference between compounds. Some uremic solutes interfere with functions that directly affect toxic action of other solutes.

## 3. Discussion

In recent years, modern analytical techniques have been developed and used in various studies to identify, characterize, and assess the biological activity of uremic toxins. Research has been directed mainly towards the resolutions of a number of problems related to evaluation of individual substances and groups of substances and their potential role in diagnosis of organ dysfunction, in particular in the kidney, cardiovascular system, and gastrointestinal system, and in neuropsychiatric disorders. There are three main sources for such compounds: exogenous ones are consumed with food, whereas endogenous ones are produced by host metabolism or by symbiotic microbiota metabolism. ASD is a neurodevelopmental disorder that likely starts to unfold in utero and is characterized by neuroplastic remodeling of the brain. ASD is, in many cases, associated with chronic gastrointestinal symptoms [8,40].

In our study, levels of five uremic toxins, pCS, IS, TMAO, SDMA, and ADMA, were measured in urine samples of ASD children and neurotypical children. We observed higher levels of pCS and IS in the ASD group compared to the controls. Moreover, the toxin levels of TMAO, SDMA, and ADMA were lower in ASD patients than in control-group patients. Similarly, for pCS and IS in children classified, according to the intensity of their symptoms, into mild, moderate, and severe, elevated levels of these compounds were observed. For mild severity of the disorder, elevated levels of TMAO and comparable levels of SDMA and ADMA for ASD children as compared to the controls were observed in urine. For moderate severity of ASD, significantly elevated levels of TMAO but reduced levels of SDMA and ADMA were observed in the urine of ASD children as compared to the controls. When the results obtained for severe ASD severity were considered, reduced levels of TMAO and comparable levels of SDMA and ADMA were observed in ASD children. These results should be approached with caution, as the group of children with significant severity was small (N = 12).

One of the compounds belonging to the group of uremic toxins is p-cresyl sulphate (pCS). pCS is host metabolites derived from bacterial metabolites originating from intestinal bacterial fermentation of tyrosine [41]. In ASD children, overgrowth of intestinal bacteria was reported. This suggests that p-cresol-producing bacteria may be present in the distal ileum. pCS is formed from p-cresol through its metabolism, which is absorption in the intestine [42]. It has been suggested that nearly 95% of p-cresol is metabolized in the body into pCS via O-sulphonation. This process occurs mainly in the liver but also in colonic epithelial cells. Several studies have pointed out increased urinary p-cresol [38,43,44] in ASD patients as compared to neurotypical controls. Two independent cohorts were conducted with ASD children from France and Italy. One initial study included 59 ASD patients with 59 sex- and age-matched neurotypical individuals [36]. ASD children under 7 years of age exhibited a 2.5-fold increase in urinary levels of p-cresol. A replication study conducted on 33 ASD patients and 33 sex- and age-matched neurotypical individuals showed a similar increase in urinary levels of p-cresol in ASD children [43]. Increased levels of p-cresol were also observed in the stool of ASD patients when compared to neurotypical individuals in two independent studies.

Urinary p-cresol levels correlate with intensity of ASD behavioral impairments, in particular stereotypies and compulsive–repetitive behaviors [36,43]. ASD behaviors were assessed using the Autism Diagnostic Observation Schedule (ADOS), the Autism Diagnostic Interview—Revised (ADI-R), the Children Autism Rating Scale (CARS) and the Vineland Adaptive Behavior Scale (VABS). Urinary p-cresol levels are also correlated with slow intestinal transit and chronic constipation in young ASD children [11].

Numerous studies have shown that pCS, a metabolic product of p-cresol, is elevated in the urine, feces or serum of children with ASD [37,39,45,46]. In an animal model, it was also shown that p-cresol intake caused social behavior deficits and repetitive behaviors through directly affecting remodeling of the gut microbiota in test mice [47,48]. In our study, we also observed elevated urinary pCS levels in children with ASD compared to the controls. In addition, they were elevated in all ASD severity groups (mild, moderate, and severe). The mechanisms responsible for action of p-cresol and its effects on social behavior in ASD have yet to be understood and elucidated.

Another compound belonging to uremic toxins, which is also a product of gut microbiota, is IS. IS’ precursor is synthesized from Trp obtained from the diet. The neurobehavioral effects of IS have been demonstrated in an animal model of rats. Studies have demonstrated the ability of IS to accumulate in brain tissues and its effects on the behavioral profile and brain monoamines. Chronic exposure to IS leads to behavioral disturbances such as decreased locomotor activity and spatial memory, as well as increased stress sensitivity. Both IS and pCS are bound to albumin in plasma. Among other things, induction of reactive oxygen species formation is indicated as a pathogenic effect of IS and pCS. Activation of the nuclear factor-kappaB (NF-κB) pathway occurs, causing oxidative stress and production of proinflammatory cytokines [49,50]. 

In chronic kidney disease (CKD), there is deterioration of renal function and an increase in symptoms of uremic syndrome. All systems and organs are affected, including the central nervous system. Among neurological disorders, uremic encephalopathy is common and manifests itself through disturbances in the emotional sphere, cognitive function, and headaches, as well as confusion or coma. Uremic toxins (UTs) induce brain dysfunction through affecting the brain’s biochemical processes and causing neurotransmitter dysfunction. Increases in neuronal calcium levels have been observed to disturb norepinephrine metabolism. Norepinephrine levels were reduced in the brains of CKD rats. There were also reductions in activity of tyrosine hydroxylase, which is involved in synthesis of norepinephrine, and monoamine oxidase, which is involved in its degradation. The release of norepinephrine into the synaptic space and its retrograde reuptake are reduced in uremia. These abnormalities may underlie behavioral disorders [51]. 

UTs accumulating in the brain can lead to elevated levels of guanidine compounds, which are responsible for cognitive deficits because their accumulation results in oxidative stress, inflammation, and neuronal damage [52,53].

Uremic toxins that can influence brain function include ADMA, SDMA, TMAO, uric acid, and urea. However, a growing body of data has indicated that there is an association between uremic toxins and disorders of the central nervous system. Furthermore, uremic toxins may predispose patients to neurological disorders. There is increasing evidence to show that uremic toxins, for example, ADMA, SDMA, TMAO, or IS, have neurotoxic effects [54].

CKD patients are often diagnosed with cardiovascular disease, including uremic cardiomyopathy. This is associated with the toxic effects of IS causing increased production of reactive oxygen species, resulting in oxidative stress, then leading to myocardial fibrosis and cardiomyocyte hypertrophy [53,55]. The neurotoxic effects of IS have also been indicated in patients with chronic kidney disease (CKD), as well as impacts on neurodegeneration in Alzheimer’s disease (AD) and other neurological disorders [49]. IS is considered a neurotoxin due to its ability to accumulate in brain tissue, mainly as a result of disruption of organic anion transporter 3 (OAT 3) function across the blood–brain barrier. It is important to note that IS can contribute significantly to disruption of CNS homeostasis and lead to neuronal damage through increased oxidative stress as well as inflammation in glial cells. It has also been suggested that the agonistic properties of the aryl hydrocarbon receptor (AhR), endothelial dysfunction, and IS-induced prothrombotic effects may also indirectly lead to neuronal damage [49].

Many patients with ASD have reported problems with the gastrointestinal tract (GIT), which may indicate involvement of the gut–brain axis in autism. It has been indicated that multiple gastrointestinal dysfunctions in ASD correlate with severity of neurological symptoms [8,56,57]. Changes in the composition of the intestinal microbiota result in consequent impaired production of specific metabolites such as uremic toxins (IS, pCS), which can be absorbed into the circulation and alter normal biochemical processes. Elevated levels of IS and pCS have been observed in ASD and may cause neuronal damage [58]. In this study, we found elevated levels of IS and pCS in the ASD group, which agrees with previous findings [30,50,59]. In research conducted by Olesova et al. (2020), IS levels in ASD patients were determined. Those researchers observed elevated levels of IS in ASD patients compared to typically developing children. All these changes were found in school-age children with ASD [50].

Similarly to IS and pCS, the TMAO precursor is also produced by intestinal bacteria. TMAO is produced through oxidation of trimethylamine (TMA) by hepatic flavin monooxygenases (FMO1 and FMO3). It is mainly formed from dietary substrates such as carnitine, phosphatidylcholine/choline, betaine, ergothioneine, and dimethylglycine, which are metabolized by the intestinal microflora in the colon. Therefore, its levels are related to the state and composition of the gut microbiome but also depend on a number of factors, such as sex, age, diet, renal function, and flavin monooxygenase activity in the liver [60]. TMAO disrupts the blood–brain barrier through reducing expression of tight junction proteins (claudin-5, tight junction protein-1), which promotes its access to the brain [61]. Reports of proinflammatory effects of TMAO in animal models and in vitro can be found in the literature, but there are no such reports in human studies. Elevated levels of TMAO have been documented in a wide variety of diseases, such as atherosclerosis, Alzheimer’s disease, and diabetes. Unfortunately, there are few reports of elevated TMA levels in ASD in the available literature. In a case–control study, Mu et al. (2019) examined TMAO levels in plasma in ASD children. They observed higher concentrations of TMAO in ASD children [62]. Quan et al. (2020) assessed plasma levels of TMAO, their association with ASD, and degree of symptom severity. They observed higher levels of TMAO in plasma in children with ASD compared to typically developing children. Elevated plasma levels of TMAO were associated with ASD symptom severity. Plasma levels of TMAO were lower in ASD children classified as mild-to-moderate (N = 66) than in ASD children whose symptoms were classified as severe (N = 98). Authors have suggested that for each 1 μmol/l increase in TMAO in plasma, there is an increased risk of severe autism by as much as 61% [63]. These results remain partly consistent with those obtained in our study. In our research, we also observed higher levels of TMAO in urine of ASD patients. Additionally, in the mild and moderate groups, TMAO levels were higher than in the severe group. However, the group of children with severe symptoms was relatively small (N = 12). In animal-model studies, TMAO supplementation was observed to cause mitochondrial impairment, synaptic damage, and cognitive impairment in the hippocampus and oxidative stress in mice [64]. Moreover, the role of TMAO in autism is not known but may reflect the changes in the gut microbiota that occur in this disorder.

ADMA and SDMA are not found naturally in the human genetic code and are therefore called nonproteinogenic amino acids. Alterations in ADMA and SDMA levels in physiological fluids are observed in the cases of many diseases, such as neurological conditions and disorders: for example, acute ischemic stroke [65], Parkinson’s disease [66], migraine [67], multiple sclerosis, and depressive symptoms [68]. The data on levels of ADMA and SDMA in ASD are scarce and somewhat inconsistent. Accumulation of uremic toxins may cause cerebral endothelial dysfunction and contribute to cognitive disorders in CKD [69].

## 4. Materials and Methods

### 4.1. Study Samples 

This study involved 191 participants. One hundred and forty-three of these participants were diagnosed with ASD. In the recruited samples, the subjects’ average age was 9.9 ± 3.5 years, in the range of 2.1–18.1 years. Forty-eight represented the general population of neurotypical children without any acute or chronic illness, who were, on average, 9.2 ± 3.9 years of age, in the range of 2.5–20.8 years. These children had no associated neurobiological syndromes. The children with ASD were not on a gluten-free, casein-free, or sugar-free diet. The demographic and clinical characteristics of the participating children and adolescents are given in Table 3. The study protocol was approved by the National Medical Ethics Committee (0120-201/2016-2 KME 78/03/16). Children in the study group were diagnosed with ASD by an expert pediatrician or a neuropsychiatrist in collaboration with a psychologist. A multidisciplinary team consisting of pediatricians, psychiatrists, and psychologists diagnosed ASD with a clinical assessment and a psychological assessment. The criteria summarized by the DSM-5 were used [70]. 

ASD clinical diagnosis relies on the Diagnostic and Statistical Manual of Mental Disorders (DSM-V) criteria [71]. DSM-V diagnostic criteria for ASD rely on evaluation of two behavioral domains: social communication impairments and restricted, repetitive behaviors [72]. When confirming ASD, it is therefore necessary to determine, in addition to the diagnosis itself, whether it is ASD with co-occurrence of intelligence impairment and speech impairment; the presence of other nervous-development, mental, and behavior disorders; or the presence of genetic and environmental factors. Due to the diversity and scope of ASD among individuals, the latter is classified into three groups according to the level of expression of deficits in social communication and according to the behavioral patterns of the individual. Deficits in social interaction are denoted by numbers from 1 to 3, with level 3 being more severe deficits in communication and interaction. Deficits in performance, interest, and activity are denoted by the letters A to C, with the first stage, A, representing easier deficits and C more difficult. Thus, children with mild deficits and fewer observed problems with integration into society are classified into category 1A and those with more pronounced problems into category 2B, and in the last category, 3C, are children classified with more severe deficits in social communication and interaction [73,74].

Clinicians follow the classification of behavior proposed by the Slovene educational authority [75,76,77], which covers impairments of social communication and interaction (assigned 1 for mild, 2 for moderate, and 3 for severe) on one hand and the presence of impairment in behavioral flexibility (assigned A for mild, B for moderate, and C for severe) on the other hand. With use of this protocol, children diagnosed with ASD were rated into three subgroups: 1: mild deficit (1A, 2A); 2: moderate deficit (1B, 2B); and 3: severe deficit (2C, 3C).

### 4.2. Analytical Method

Reference standards, i.e., ADMA, SDMA, and TMAO, as well as internal standards ADMA-d6 and TMAO-d9 were purchased from Toronto Research Chemicals (Toronto, Canada). 

Uremic toxins were determined with validated high-performance liquid chromatography coupled to mass spectrometry (LC-MS/MS) using multiple-reaction-monitoring (MRM) mode on Agilent 1260 Infinity (Agilent Technologies, Santa Clara, CA, USA) coupled to QTRAP 4000 (AB Sciex, Framingham, MA, USA). MRM transitions, declustering potential (DP), and collision energy (CE) were: ADMA, m/z 203 > 46 (DP = 61V, CE = 41V); SDMA, m/z 203 > 172 (DP = 61V, CE = 19V); ADMA-d6, m/z 209 > 77 (DP = 66V, CE = 45V); TMAO, m/z 76 > 42 (DP = 66V, CE = 53V); and TMAO-d9, m/z 85 > 46 (DP = 61V, CE = 59V), respectively. Chromatographic separation was achieved using a SeQuant^®^ ZIC^®^-HILIC (50 × 2.1 mm, 5 μm, Merck (Darmstadt, Germany)) column. The column was maintained at 25 °C at a flow rate of 0.5 mL min^−1^. The mobile phases consisted of 20 mM ammonium acetate as eluent A and acetonitrile with 0.2% formic acid as eluent B. The gradient (%B) was as follows: 0 min, 95%; 1 min, 95%; 7 min, 50%; and 8 min, 50%. The injection volume was 5 μL. Urine samples (0.1 mL) prior to injection with LC were mixed with internal standards (0.1 mL, 6 μg/mL) and acetonitrile (0.6 mL), vortexed at high speed (3 min), and centrifuged (5 min at 10,000 g).

### 4.3. Statistical Analysis

All statistical analyses were performed using the R programming language. Differences between the groups were determined statistically using a one-way analysis of variance (ANOVA) followed by the Bonferroni multiple-comparisons method and Student’s t-test. Differences at *p*-values < 0.05 were taken as significant statistically.

## 5. Strengths and Weaknesses of Our Study

The strengths of this study are that we did not find a comparable study according to the criteria of separation between high-functioning autism (F84.5) and autism (F84.0), we did not find any study indicating a difference in the case of additional diagnoses, and we also did not find any difference according to level of presenting disorder (mild, moderate, and severe). The weaknesses of our study are the proportion of the control group versus the group diagnosed with ASD and the disproportionate female versus male occupancy in the patient group. However, as our calculations showed, there were no statistically significant gender differences in our cohort.

## 6. Conclusions

Determination of metabolites in urine gives the opportunity for a relatively easy and fast diagnosis of gastrointestinal disorders, which are one of the most common ailments in children with ASD. The results of these determinations can also be used to help guide and evaluate effectiveness of treatment. Understanding the trends of concentrations of compounds such as the uremic toxins that are discussed in this publication—pCS, IS, TMAO, SDMA, and ADMA—also makes it possible to link them with the severity of neurological disorders. In the ASD group, significantly higher levels of pCS and IS concentrations and lower levels of TMAO, SDMA, and ADMA concentrations were observed than in the control group. For mild severity of the disorder, increased levels of TMAO concentration and comparable levels of SDMA and ADMA concentrations were observed in children with ASD when compared to the control group. For moderate ASD severity, significantly increased levels of TMAO concentration but lower levels of SDMA and ADMA were observed. However, for severe ASD severity, reduced levels of TMAO concentration and comparable levels of SDMA and ADMA were observed in children with ASD. The discussed results will serve as a base for further, more detailed and expanded research.

## Figures and Tables

**Figure 1 ijms-24-07078-f001:**
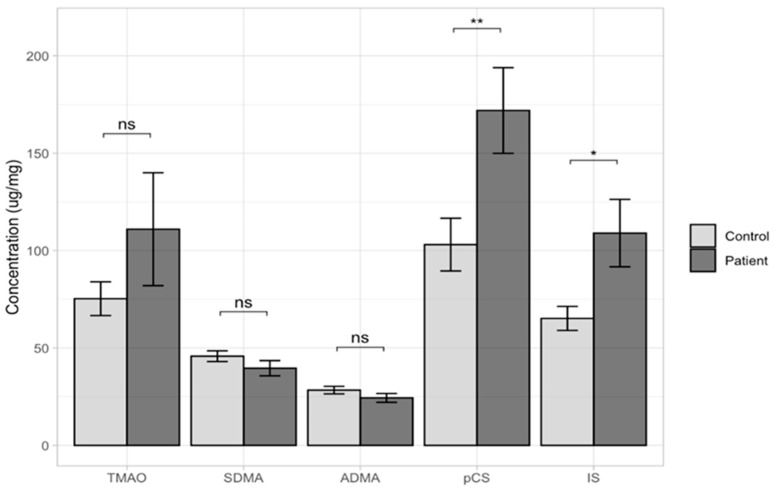
Determined levels of all uremic toxins in the study groups (ASD + HFA patients and controls). ns—not significant difference; *—statistically significant, with p < 0.05; **—statistically significant, with *p* < 0.001.

**Figure 2 ijms-24-07078-f002:**
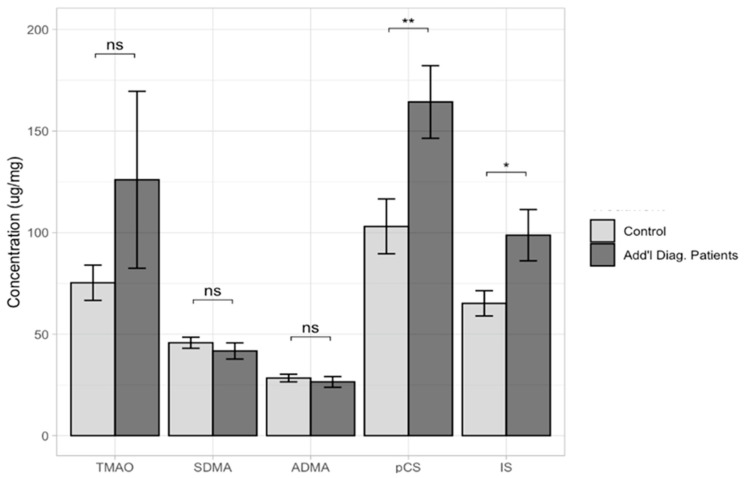
Determined levels of all uremic toxins in the ‘additional diagnosis’ patients’ group vs. the control group. ns—not significant difference; *—statistically significant, with p < 0.05; **—statistically significant, with *p* < 0.001.

**Figure 3 ijms-24-07078-f003:**
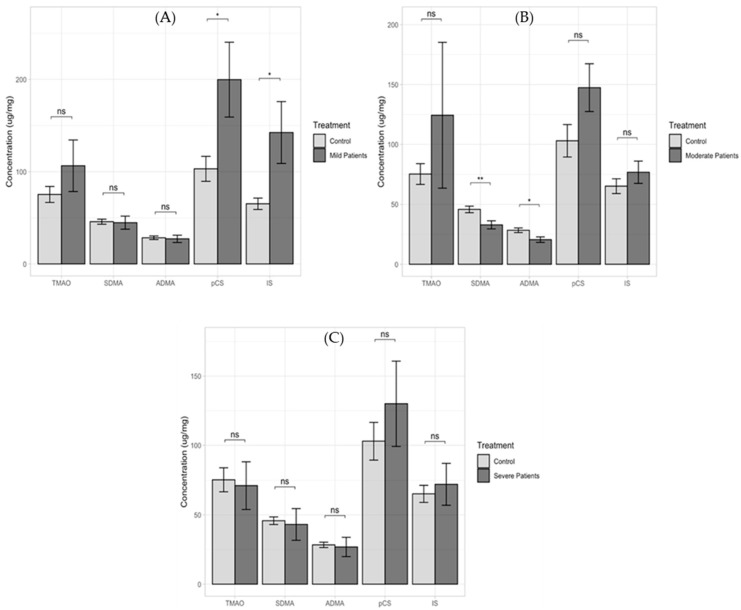
Comparison of levels of TMAO, SDMA, ADMA, pCS, and IS (**A**) in a group of mild-classified patients vs. the control group (71 vs. 48); (**B**) in a group of moderate-classified patients vs. the control group (60 vs. 48); and (**C**) in a group of severe-classified patients vs. the control group (12 vs. 48). ns—not significant difference; *—statistically significant, with p < 0.05; **—statistically significant, with *p* < 0.001.

**Table 1 ijms-24-07078-t001:** Levels of uremic toxins in urine samples in study groups (ASD, N = 143; controls, N = 48).

Unit [µg/mg]	Min	Max	Media	SD	RSD	25–75 P	Normal Distr.
TMAO	3.14	3689.53	53.96	346.49	3.12	33.56–81.70	<0.0001
c–TMAO	15.53	296.80	56.94	59.99	0.79	32.71–103.30	<0.0001
ADMA	1.01	179.64	15.93	27.05	1.10	10.38–27.41	<0.0001
c–ADMA	7.07	69.24	24.62	13.40	0.47	19.89–31.34	0.0003
SDMA	2.06	360.46	26.37	46.81	1.18	18.16–41.95	<0.0001
c–SDMA	15.44	100.31	38.95	19.05	0.41	33.19–58.67	0.0008
IS	3.15	1689.66	55.64 *	206.87	1.89	32.76–112.09	<0.0001
c–IS	4.37	211.55	52.62	42.67	0.65	35.33–86.15	0.0002
pCS	0.37	2161.24	93.68 *	262.81	1.52	41.08–206.85	<0.0001
c–pCS	1.29	398.93	73.99	93.75	0.90	39.82–117.80	<0.0001

c—control; *—statistically significant, with *p* < 0.05.

**Table 2 ijms-24-07078-t002:** Nonparametric percentile method (CLSI C28-A3).

Compound	Lower Limit	Upper Limit
TMAO	15.84	285.47
SDMA	16.85	97.09
ADMA	7.97	66.07
IS	6.28	198.91
pCS	1.77	382.35

**Table 3 ijms-24-07078-t003:** Description of patients and controls used in this study.

	Patients	Controls
N	143	48
Female	17	25
Male	126	23
Age		
Average	9.9 ± 3.5	9.2 ± 3.9
Under 8 Years	46	20
Above 8 Years	97	28
Classification 1		48
Mild	71	
Moderate	60	
Severe	12	
Classification 2		48
ASD ^1^	105	
HFA ^2^	38	
Additional Diagnosis		48
No	57	
Yes ^3^	86	

^1^ ASD—autism spectrum disorder, ^2^ HFA—high-functioning autism, ^3^ Additional diagnoses: G40—epilepsy and recurrent seizures; H91—other and unspecified hearing loss; R56—convulsions, not elsewhere classified; R62.0—delayed milestones in childhood; F81—specific developmental disorders; F90—attention-deficit hyperactivity disorder; F95—tic disorder. 4.2. Neuropsychiatric assessment and behavior classification.

## Data Availability

The data presented in this study are available on request from the corresponding authors.

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
