# Peer review of "Relationship between Excreted Uremic Toxins and Degree of Disorder of Children with ASD"

_ijms, 2023, doi:10.3390/ijms24087078_

Round 1

Reviewer 1 Report

Autism spectrum disorders (ASD) are a diverse group of conditions. They are characterised by some degree of difficulty with social interaction and communication. Other characteristics are atypical patterns of activities and behaviours, such as difficulty with transition from one activity to another, a focus on details and unusual reactions to sensations. It is estimated that worldwide about one in 100 children has autism.

The role of uremic toxins in the development of the ASD is intensively discussed recently, however, relations are not clearly elucidated.

The article Relationship between excreted uremic toxins and the degree of 2 disorders of children with ASD written by Osredkar at al is well written article on the actual topic. Introduction contains all necessary background regarding presented topic. The aim of the study is clear and well formulated. Results and Methodology are described sufficiently. Discussion is exhausting regarding the topic.

The article is suitable for the publication after minor changes based on my comments. Please, see my comments below.

1)      Study group consists from 143 patients and 48 controls. Patient group consists from 17 females and 126 males.  The numbers of probands in individual files vary greatly, which can cause an imbalance in the statistical evaluation. Also the stratification under 8 years and above 8 years results into two groups with the very different number of probands. Describe if you reflect these facts during statistical evaluation. This imbalance in the number of probands into the subgroups should be mentioned between strengths and weaknesses of the study.

2)      Strengths and weaknesses of the study are missing. In connection with the point one of my evaluation, strengths and weaknesses should be mentioned before the Conclusion.

3)      Table 1 which is presented at the line 118 in the part Results is in fact summary of the Study group and Study samples. Table 1 with appropriate information should be shifted to the part 4. Material and Methods.

4)      Add Study groups as thge first part of the part 4. Material and Methods, then continue with the part 4.2. Study samples or merge parts Study group and Study samples and place the Table 1 here.  

Author Response

The authors are appreciating you for your precious time in reviewing our paper and providing valuable comments. It was your valuable and insightful comments that led to possible improvements in the current version. The authors have carefully considered the comments and tried our best to address every one of them. We hope the manuscript after careful revisions meets your high standards. In the revised version of the article, all the corrections relating to the suggestions have been made. Please let us know if you still have any questions or concerns about the manuscript. We will be happy to address them.

1. Study group consists from 143 patients and 48 controls. Patient group consists from 17 females and 126 males.  The numbers of probands in individual files vary greatly, which can cause an imbalance in the statistical evaluation. Also the stratification under 8 years and above 8 years results into two groups with the very different number of probands. Describe if you reflect these facts during statistical evaluation. This imbalance in the number of probands into the subgroups should be mentioned between strengths and weaknesses of the study.

The authors thank you for this comment. We are aware of this and our reply follows below.

Our sampling was carried out at the University Medical Centre Ljubljana in the autism outpatient clinic. The doctor had the patients scheduled in random order and presented our study to the parents and invited them to participate. Those who agreed signed an informed consent and the child provided the urine sample. During this time, we did not refuse anyone who wanted to be in the study and this is how we got the inclusion ratio. This disproportionality has of course been carried over to the age distribution.

When the results were statistically processed, we checked the results by sex and found no statistically significant differences.

Calculations with corresponding graphs have been uploaded in Supplementary Materials file.

Comparison of the statistical calculations shows that the values of the individual parameters are not statistically different, but there is a large scatter in the results. We have described this situation in the text in the strengths and weaknesses section of the study.

2. Strengths and weaknesses of the study are missing. In connection with the point one of my evaluation, strengths and weaknesses should be mentioned before the Conclusion.

A section on the strengths and weaknesses of the study has been added to the text, as here:

The strengths of the study are that we did not find a comparable study according to the criterion of separation between high-functioning autism (F84.5) and autism (F84.0), we did not find a study indicating a difference in the case of additional diagnoses, and we also did not find a difference according to the level of the presenting disorder (mild, moderate and severe). The weakness of our study is the proportion of the control group versus the group diagnosed with ASD, and the disproportionate female versus male occupancy in the patient group. However, as our calculations showed, there are no statistically significant gender differences in our cohort.

3. Table 1 which is presented at the line 118 in the part Results is in fact summary of the Study group and Study samples. Table 1 with appropriate information should be shifted to the part 4. Material and Methods.

Thank you for this comment. We moved table 1 into section 4 and renumbered it.

4. Add Study groups as thge first part of the part 4. Material and Methods, then continue with the part 4.2. Study samples or merge parts Study group and Study samples and place the Table 1 here.  

As point 3).

Reviewer 2 Report

The authors have submitted a research article regarding a possible association of severity of autism spectrum disorder (ASD) with the urine levels of uremic toxins.

For this purpose, the authors have investigated levels of some uremic toxins by evaluating their urine concentrations using an HPLC-MS, illustrating a hypothesis suggesting that, in the aspect of ASD, high levels of uremic toxins such as pCS and IS are observed in patients with a severe ASD. The authors discussed the beneficial availability of the urine levels of uremic toxins which may be recognized to be biomarkers for ASD, resulting in reliable perspectives. This issue is of interest, and impact of their results is strong. My overall concern with the article describing the current available data regarding beneficial availability of some possible biomarkers which are available for evaluation of ASD, offer something substantial that helps advance our understanding of effective management which draws novel class of effective compounds available in clinic.

To strengthen authors’ perspectives, the authors are strongly recommended to add a “relationship” discussion in detail regarding known uremic toxin effect on humans (not only chronic kidney disease but also neurological disorders), for instance.

Author Response

The authors are appreciating you for your precious time in reviewing our paper and providing valuable comments. It was your valuable and insightful comments that led to possible improvements in the current version. The authors have carefully considered the comments and tried our best to address every one of them. We hope the manuscript after careful revisions meets your high standards. In the revised version of the article, all the corrections relating to the suggestions have been made. Please let us know if you still have any questions or concerns about the manuscript. We will be happy to address them.

The authors have submitted a research article regarding a possible association of severity of autism spectrum disorder (ASD) with the urine levels of uremic toxins.

For this purpose, the authors have investigated levels of some uremic toxins by evaluating their urine concentrations using an HPLC-MS, illustrating a hypothesis suggesting that, in the aspect of ASD, high levels of uremic toxins such as pCS and IS are observed in patients with a severe ASD. The authors discussed the beneficial availability of the urine levels of uremic toxins which may be recognized to be biomarkers for ASD, resulting in reliable perspectives. This issue is of interest, and impact of their results is strong. My overall concern with the article describing the current available data regarding beneficial availability of some possible biomarkers which are available for evaluation of ASD, offer something substantial that helps advance our understanding of effective management which draws novel class of effective compounds available in clinic.

To strengthen authors’ perspectives, the authors are strongly recommended to add a “relationship” discussion in detail regarding known uremic toxin effect on humans (not only chronic kidney disease but also neurological disorders), for instance.

Thank you for your all valuable suggestions. The recommended information in accordance with the Reviewer's comment was added at lines 256-280;

In chronic kidney disease (CKD), there is a deterioration of renal function and an increase in the symptoms of uremic syndrome. All systems and organs are affected, including the central nervous system. Among the neurological disorders, uremic encephalopathy is common and manifests itself by disturbances in the emotional sphere, cognitive function, headaches, as well as confusion or coma. Uremic toxins (UTs) induce brain dysfunction by affecting its biochemical processes and causing neurotransmitter dysfunction. An increase in neuronal calcium levels is observed to disturb norepinephrine metabolism. Its level was reduced in the brain of CKD rats. There was also a reduction in the activity of tyrosine hydroxylase, which is involved in the synthesis of norepinephrine, and monoamine oxidase, which is involved in its degradation. The release of norepinephrine into the synaptic space and its retrograde reuptake are reduced in uremia. These abnormalities may underlie behavioural disorders [52].

UTs accumulating in the brain can lead to elevated levels of guanidine compounds, which are responsible for cognitive deficits because their accumulation results in oxidative stress, inflammation and neuronal damage [53,54].

Uremic toxins that can influence brain function include ADMA, SDMA, TMAO, uric acid and urea. However, a growing body of data indicates that there is an association between uremic toxins and disorders of the central nervous system. Furthermore, uremic toxins may predispose patients to neurological disorders. There is increasing evidence to show that uremic toxins, for example ADMA, SDMA, TMAO or IS, have a neurotoxic effect [55].

CKD patients are often diagnosed with cardiovascular disease, including uremic cardiomyopathy. This is associated with the toxic effects of IS causing increased production of reactive oxygen species resulting in oxidative stress, leading to myocardial fibrosis and cardiomyocyte hypertrophy [54,56]. 

Round 2

Reviewer 2 Report

The authors have addressed properly all the issues raised by reviewers including me. I have no more comments, and now recommend that this manuscript is acceptable for publication in the journal IJMS.